# Ribosome and Translational Control in Stem Cells

**DOI:** 10.3390/cells9020497

**Published:** 2020-02-21

**Authors:** Mathieu Gabut, Fleur Bourdelais, Sébastien Durand

**Affiliations:** 1Equipe ‘Transcriptome Diversity in Stem Cells’, Cancer Cell Plasticity Department, INSERM 1052, CNRS 5286, Cancer Research Center of Lyon, Centre Léon Bérard, 69008 Lyon, Francefleur.bourdelais@gmail.com (F.B.); 2Université Claude Bernard Lyon 1, 69100 Villeurbanne, France

**Keywords:** ribosomes, stem cells, translational regulation, ribosomal proteins, rRNA modifications, ribosome biogenesis, specialized ribosomes

## Abstract

Embryonic stem cells (ESCs) and adult stem cells (ASCs) possess the remarkable capacity to self-renew while remaining poised to differentiate into multiple progenies in the context of a rapidly developing embryo or in steady-state tissues, respectively. This ability is controlled by complex genetic programs, which are dynamically orchestrated at different steps of gene expression, including chromatin remodeling, mRNA transcription, processing, and stability. In addition to maintaining stem cell homeostasis, these molecular processes need to be rapidly rewired to coordinate complex physiological modifications required to redirect cell fate in response to environmental clues, such as differentiation signals or tissue injuries. Although chromatin remodeling and mRNA expression have been extensively studied in stem cells, accumulating evidence suggests that stem cell transcriptomes and proteomes are poorly correlated and that stem cell properties require finely tuned protein synthesis. In addition, many studies have shown that the biogenesis of the translation machinery, the ribosome, is decisive for sustaining ESC and ASC properties. Therefore, these observations emphasize the importance of translational control in stem cell homeostasis and fate decisions. In this review, we will provide the most recent literature describing how ribosome biogenesis and translational control regulate stem cell functions and are crucial for accommodating proteome remodeling in response to changes in stem cell fate.

## 1. Introduction

The development of high-throughput methods for studying gene expression has considerably improved our understanding of cellular regulatory networks and our insight into how a multitude of molecular processes are coordinated to adapt the proteome to specific cell functions and respond accurately to fluctuating environmental cues. These multi-omics approaches have allowed researchers to investigate large-scale changes in epigenetic status, chromatin structure, as well as RNA expression and stability. The fields of stem cell and developmental biology have also benefited from these technological breakthroughs. Many studies have characterized epigenetic regulations of chromatin structure in stem cells and during early embryonic development. RNA expression has been extensively explored in non-synchronized cell cultures but also at the single-cell level, either from in vitro models or dissociated tissues, therefore providing the scientific community with precious resources. However, multiple studies suggested that messenger RNA (mRNA) levels are poorly correlated with protein levels [1,2,3]. Indeed, it is estimated that only 40% of changes in protein abundance are attributable to related changes in RNA levels [3]. Similar observations have been drawn in embryonic stem cells (ESCs) and induced pluripotent stem cells (iPSCs), the proteome of which is also poorly correlated with its transcriptome [4]. Some groups reported that changes in protein expression observed during ESC and adult stem cell (ASC) differentiation could primarily result from changes in translation, with a minor contribution from variations in RNA levels, therefore highlighting the key role of translational regulation in rapid cell identity changes [5,6]. These observations emphasize the requirement for systematic cellular proteome analysis to study gene expression regulation or alternatively the need for methodologies that globally measure mRNA translation status, such as ribosome profiling. Therefore, a comprehensive understanding of complex changes in gene expression requires a global analysis of the coordinated interplay between epigenetic, transcriptional, and translational mechanisms. This integrated vision of gene expression is increasingly accessible owing to innovative sequencing technologies (e.g., Ribo-Seq, RiboMet-seq), as well as adapted computational analysis methods and improved calculation capacities.

During the past decade, the translation process has been thoroughly investigated in stem cells and increasing evidence directly links regulations of translation and of ribosome biogenesis to the control of stem cell homeostasis and cell fate. In this review, we will discuss current knowledge on (i) the global regulation of translation in stem cells and its role during cell fate transition, on (ii) the contribution of ribosome biogenesis to the control of stem cell homeostasis, and (iii) we will finally present new emerging concepts proposing that ribosomes, largely considered to be devoid of regulatory activities so far, could directly control specific gene expression programs and potentially impact stem cell biology and early embryonic development.

## 2. Control of Translation in Stem Cells

### 2.1. Global Translation Increases During Stem Cell Differentiation

ESCs represent an interesting model to study gene expression due to an atypical molecular identity. Although they possess a relatively open chromatin structure and a highly permissive transcription, ESCs must establish and maintain the expression of specific pluripotent genetic programs while concomitantly silencing lineage-specific genes. In addition, ESCs need to maintain the capacity to respond to cues inducing changes in cell identity by quickly rewiring gene expression programs. Hence, the translation process, which occurs downstream of most mRNA maturation steps, could indeed rapidly regulate protein synthesis without demanding complex modulations of epigenetic and RNA expression pathways, and may therefore be a crucial regulatory step governing gene expression programs required for both stem cell identity and transitions in cell fate (Figure 1) [6,7,8]. 

To examine the role of translation in cell identity control, Sampath and colleagues [10] investigated the regulation of translation during ESC fate decisions. They observed that, unlike undifferentiated ESCs, embryoid bodies (EBs) differentiated from ESCs displayed cellular structural features characteristic of increased translational efficiency, such as more abundant Golgi bodies and rough endoplasmic reticulum (ER) contents, and a larger cytoplasmic volume. Accordingly, ^35^S-methionine labeling clearly demonstrated that translational efficiency is two-fold lower in undifferentiated ESCs compared to differentiated EBs. In addition, the polysome density (i.e., the fraction of actively translating ribosomes) was increased by 60% in EBs compared to undifferentiated ESCs, suggesting that mRNAs are more actively translated in EBs. This observation that translational efficiency is lower in ESCs than in differentiated progenies has been supported by other reports [11,12,13]. Interestingly, decreasing translational efficiency during somatic cell reprogramming enhances iPSC production [14,15]. Of note, a contradictory observation that translational efficiency diminishes during ESC differentiation upon retinoic acid (RA) treatment has also been reported [16]. While it is unclear what causes this discrepancy, RA treatment employed to induce differentiation might incidentally directly inhibit translation as previously suggested [17,18,19]. 

Surprisingly, although global translation and polysome mRNA loading are low in ESCs, several studies indicate that ESCs maintain high levels of free albeit inactive ribosomes, at the expense of a tremendous energy cost ([10,16,20], our observations). As proteome rewiring during ESC differentiation is mainly regulated at the translational level [5,6], maintaining an elevated pool of available ribosomes might be essential for rapidly remodeling and priming gene expression towards differentiation programs. Accordingly, ribosome production is highly regulated in stem cells and undoubtably contributes to stem cell homeostasis as we will discuss in the third and fourth parts of this review. 

Interestingly, low translational efficiency also seems to be a common feature of ASCs and has been repetitively reported in hematopoietic, neural, muscle, skin, and germline stem cells [21,22,23,24,25,26,27,28]. Consistently, Signer and colleagues [25] showed that translational efficiency is lower in hematopoietic stem cells (HSCs) compared to restricted hematopoietic progenitors. Interestingly, increasing HSC proliferation does not compensate for a lower translation rate. In addition, hair follicle stem cells (HFSCs) display a lower translation rate than committed and differentiated cells [21]. Similar to HSCs, translation is uncoupled from proliferation as non-cycling (G_0_/G_1_) committed stem cells possess a higher translational efficiency than proliferating (S/G_2_/M) HFSCs. Finally, in a mouse model of skin squamous tumors, proliferative tumor-initiating cells (also referred to as cancer stem cells) exhibited a reduced translational activity compared with differentiated cancer cells, independently of the proliferative status [21]. Altogether, this suggests that the translation rate is correlated with the differentiation and commitment statuses of stem cells rather than with their intrinsic proliferative capacity.

### 2.2. The Global Baseline of Translation Is Tightly Regulated in Stem Cells

Even though protein synthesis is lower in stem cells, maintaining a specific translational baseline is crucial, as alterations that enhance or lower translational efficiency are detrimental to stem cell homeostasis. Indeed, forcing an increase in translation drives human ESCs towards differentiation [11]. In addition, knockout of the pseudouridine synthetase PUS7 gene in ESCs, which increases global translation by preventing the formation of translation inhibiting tRNA fragments, promotes abnormal expression of lineage-specific markers and therefore causes defects in ESC differentiation, more specifically toward the mesoderm lineage [13]. In contrast, maintaining a basal translational level is required for establishing ESC-specific chromatin structures since translation inhibition profoundly perturbs ESC euchromatin organization and permissive transcription [29]. Thus, acute translation inhibition depletes euchromatin histone marks (H3/H4 acetylation) and induces a global decrease (90%) in RNA polymerase (Pol) I/II transcripts. Interestingly, decreasing translation also deactivates developmental enhancers, suggesting that translation inhibition may impede ESC differentiation [29]. Translation inhibition by depletion of SSUP (“Small Subunit Processome”) components or the use of the translation inhibitor 4EGI-1 also alters ESC identity [16]. Thus, both translation inhibition and overactivation disrupt ESC homeostasis. 

This Janus-like characteristic of translational requirements in ESCs has also been documented in ASCs. Hence, a depletion of the RNA methyltransferase NSUN2, which reduces global translation, strengthens HFSC quiescence and promotes aberrant stem cell differentiation [30,31,32]. Similarly to the phenotype observed in ESCs, elevated global translation caused by a depletion of the pseudouridine synthetase PUS7 severely impacts HSC differentiation into committed progenitors [13]. Interestingly, Signer and colleagues [25] reported that both an increase and a decrease in translation alter HSC homeostasis. Indeed, a hypomorphic Rpl24 deletion (Rpl24−/+) provoked a 30% decrease in global translation and impaired HSC functions, presumably by affecting HSC proliferation. In contrast, complete deletion of PTEN, which therefore activated the AKT/mTOR pathways and resulted in a 30% increase in protein synthesis, also abolished HSC functions. More importantly, the hypomorphic Rpl24 deletion in PTEN-deficient HSCs brought translation rates back to lower levels and almost completely rescued HCS functions. Therefore, translation needs to be precisely balanced and regulated to sustain HSC functions in vivo [25].

### 2.3. Mechanisms Regulating Translation in Stem Cells

As described above, global translation in undifferentiated stem cells is generally maintained at a low level and needs to be tightly regulated. However, despite this dampening of global translational activity, stem cells need to sustain a proper expression level of key stemness factors in order to maintain their specific identities [7,10,11,12,29,33]. Conversely, the expression of these factors must be repressed during differentiation while global protein synthesis increases. We will now present some literature providing mechanistic insights on how stem cells regulate global translation while uncoupling peculiar translation of key stemness-encoding transcripts (summarized in Figure 1). 

Regulations of translation have been extensively described in excellent reviews, and we will thus only highlight the general principles of this process (for a review, see [34,35,36]). Translation is mainly regulated at the initiation step, although there are examples of regulation affecting elongation and termination. Translation initiation starts with the highly coordinated assembly of initiation factors (eIFs) at the 5’ end of mRNAs. The mRNA tri-methylated-guanosine cap directly interacts with eIF4E1 (referred as eIF4E throughout), which assembles with eIF4G and the RNA helicase eIF4A to form the eIF4F complex. Cap-bound eIF4F together with eIF4B and the polyA binding protein (PABP) recruit the 43S pre-initiation complex (PIC), which is composed of the ternary complex GTP-bound eIF2 associated with the methionyl-initiator tRNA (met-tRNA), the small 40S ribosomal subunit as well as eIF3, eIF5, eIF1A, and eIF1 to form the 48S complex. The PIC complex scans mRNAs from the 5’ end through the 5’ untranslated region (UTR) until it detects the AUG start codon via sequence complementarity with the met-tRNA anticodon. The RNA helicase activity of eIF4A ensures an efficient 5’UTR scanning by resolving complex secondary RNA structures. The bona fide start codon usually consists in the first AUG codon within the proper Kozak consensus (GCCRCCAUGG) context. eIF2-bound GTP hydrolysis stabilizes this 48S complex at the AUG position. Subsequent eIF2-GDP release induces the loss of most eIFs and therefore allows the recruitment of GTP-eIF5B and of the 60S large subunit to form the translation competent 80S ribosome. eIF5B-bound GTP hydrolysis releases eIF1A, therefore freeing the ribosome amino-acyl-tRNA acceptor site (A site) and allowing ribosome translocation and translation elongation. Unlike the cap-dependent eIF4E-mediated translation, assembly of initiation complexes can occur independently of the mRNA cap through specific sequences within 5’UTR called internal ribosome entry sites (IRES).

#### 2.3.1. 5’UTR Structures

Start codon selection is a very flexible process, which may not only produce different protein isoforms with distinct N-termini but also actively participates in the quantitative regulation of protein synthesis. Thus, AUG within a poor Kozak sequence context can undergo “leaky scanning” events, which promote the use of a downstream AUG within a more favorable context. Therefore, if translation is not interrupted between these two AUGs, i.e., there is no stop codon, different protein isoforms may be translated, with distinct N-termini or different primary structures (if AUGs specify shifted reading frames). Alternatively, if the upstream AUG is separated from the downstream AUG by a stop codon, it creates an upstream open reading frame (uORF) different from the main protein-encoding ORF (mORF). Notably, these uORFs are important regulators of translation and RNA stability and are present in 50% of mammalian mRNAs. Hence, uORFs capture scanning PICs and therefore impede translation of downstream mORFs or target mRNAs for nonsense-mediated decay [37,38,39,40]. In contrast, uORF translation can also stimulate downstream initiation and enhance mORF translation [40]. Regulated leaky scanning of uORFs therefore controls translation of mORFs. Of note, non-AUG start codons that differ from canonical AUGs by a single nucleotide (near-cognate triplets) can be used to initiate translation but rely on a strong Kozak context. These near-cognate triplets can also define regulatory uORFs [7].

Interestingly, AUG and non-AUG uORFs are commonly used in undifferentiated ESCs but less frequently in differentiated EBs [7]. In addition, these uORFs have regulatory potentials and may be crucial for the translational regulation of specific transcripts while global translation is maintained at a low level. Indeed, NANOG and C-MYC mRNAs, which encode key pluripotency factors, contain multiple uORFs that are used in ESCs and stimulate mORF translation [7,12]. In addition, key stemness factors of muscle stem cells (MuSCs) also rely on uORFs to regulate their expression [28]. In ESCs and MuSCs, the translation of uORF-regulated stemness factors requires an inefficient global translation, emphasizing the ability of these cells to promote translation of peculiar transcripts while repressing global translation.

Similarly, IRES-mediated translation may regulate ESC homeostasis by promoting translation of specific factors while cap-dependent global translational efficiency is low [41]. Thus, the eIF4G homolog eIF4G2, also known as DAP5, has been described as a regulator of cap-independent translation. eIF4G2, which is devoid of eIF4E-binding and PABP-binding abilities, can assemble initiation complexes on IRES-containing mRNAs and therefore initiates translation independently of the mRNA cap and eIF4E availability [42,43,44,45]. Yoffe and colleagues [41] demonstrated that eIF4G2 is required for human ESC differentiation. Interestingly, eIF4G2 regulates the expression of many genes linked to differentiation, such as the epigenetic regulator HMGN3, but also of genes encoding ribosomal proteins (RPs). The authors proposed that the effects of eIF4G2 depletion on ESC differentiation were at least in part caused by a decrease in HMGN3 expression, the mRNA translation of which is controlled by an eIF4G2-regulated IRES. Thus, 5’UTR structures are important contributors to the regulation of translation that control ESC homeostasis and changes in cell identity.

#### 2.3.2. The mTOR Pathway

Protein synthesis uses a lot of energy to sustain cell homeostasis and support cell growth and proliferation. Therefore, many translational regulatory pathways sense extracellular cues, such as nutrient availability or growth signals, as well as intracellular cues, such as metabolites or stress, to adapt protein synthesis to cellular status. Among these, the mTOR (mechanistic target of rapamycin) pathway is a master regulator of cell energy homeostasis that integrates many inputs from intra- and extracellular cues to precisely regulate the efficiency of translation (for a review, see [46,47]). The mTOR Ser/Thr kinase forms two distinct complexes, mTORC1 and mTORC2, which possess specific molecular functions and are differentially regulated. Unlike mTORC2, mTORC1 regulates translation and is composed of the catalytic mTOR kinase, mLST8 (mammalian Lethal with Sec-13 protein 8), DEPTOR (DEP domain-containing mTOR-interacting protein), the Tti/Tel2 complex, RAPTOR (regulatory-associated protein of mTOR), and PRAS40 (proline-rich Akt substrate 40kDa). mTORC1 activity is negatively regulated by the heterodimer TSC1/TSC2 (tuberous sclerosis 1 and 2). Hence, both PI3K/AKT and RAS signaling pathways suppress the activity of the TSC1/2 complex and promote mTORC1 activation.

mTORC1 regulates translation at both initiation and elongation steps. eIF4E activity is regulated by 4EBP1/2 (4EBPs), which tightly binds eIF4E and sequesters it away from eIF4G and eIF4A, therefore inhibiting the formation of eIF4F. mTORC1 phosphorylates 4EBPs and prevents them from interacting with eIF4E, thereby stimulating eIF4F complex formation and cap-dependent translation. mTORC1 also activates S6 kinases (S6Ks, S6K1/2) that stimulates translation at different levels. First, S6Ks phosphorylate and activate eIF4B, stimulating the recruitment of the 43S PIC and mRNA scanning. Second, S6Ks inhibits eEF2 kinase (eEF2K), preventing eEF2 inactivation and simulating elongation. Finally, S6Ks phosphorylate the ribosomal protein RPS6, the role of which is poorly described. However, it has been proposed that phosphorylated RPS6 might play a role in the translational regulation of mRNAs harboring a “5’ Terminal OligoPyrimidine tract” (TOP) as we will discuss in Section 3.5.

The mTOR pathway is a crucial regulator of stemness and differentiation (Figure 1). In ESCs, repressing the mTOR pathway is crucial to maintain their homeostasis [10,11,48]. Sampath and colleagues [10] demonstrated that 4EBP1 is hypo-phosphorylated in undifferentiated ESCs, suggesting that 4EBP1 actively represses cap-dependent translation in pluripotent cells. ESC differentiation into EBs promotes mTORC1 activation and therefore causes phosphorylation-dependent inhibition of 4EBP1. Similarly, Easley et al. [11] found that ESCs display a high level of mTORC1-inhibiting TSC1/2 complexes. Moreover, S6K1 activation by either TSC2 depletion or constitutively active S6K1 mutants induces ESC differentiation, therefore demonstrating that mTOR repression is required to maintain ESC pluripotency. Interestingly, mTOR pathway activation during early embryogenesis is important for developmental timing, as its partial inhibition causes blastocyst development pause (diapause) [48]. Repression of the mTOR pathway has also been reported in somatic stem cells [22,24,25,26]. Active hypo-phosphorylated 4EBPs are required for neural stem cell (NSC) and HSC functions while inactivation of 4EBPs by mTORC1-mediated phosphorylation is required for differentiation [22,26]. Finally, inhibition of the mTOR pathway by rapamycin seems to either improve or decrease somatic cell reprogramming [14,15]. While the cause of these opposite effects induced by rapamycin remains unclear, there is evidence that the mTOR pathway is dynamically regulated throughout the somatic cell reprogramming process and plays a major role in reprogramming [49].

#### 2.3.3. The Mitogen-Activated Protein Kinase (MEK) Pathway

The MEK pathway is another major regulator of translation, which adjusts translational efficiency to growth, survival, and proliferation. Hence, MEK activates ERK (extracellular signal-regulated kinase) and p38 MAPK (mitogen-activated protein kinase), which subsequently phosphorylates and activates MNK1/2 (MAPK-interacting kinases). MNK1/2 also activates translation at the initiation step by binding to eIF4F and phosphorylating eIF4E. Although the exact mechanism is unclear, eIF4E phosphorylation by MNKs is associated with an increase in translational efficiency. 

As previously described, 4EBP-dependent translational repression promotes NSC self-renewal and prevents differentiation [22]. However, it has been reported that MNK-mediated eIF4E phosphorylation is elevated in self-renewing neural progenitor cells (NPCs) and lost upon differentiation, in contrast to mTORC1 activity [33]. The effects of eIF4E phosphorylation on global translation in NPCs were not addressed but seem to strongly promote the expression of KDN5A, which is crucial for maintaining NPC stemness. This suggests that MNK-regulated translation is stimulated in NPCs and may regulate the expression of specific stemness-promoting factors.

To our knowledge, MNK activity and eIF4E phosphorylation have so far not been thoroughly investigated in ESCs. Mouse ESCs are routinely cultured in media containing serum and leukemia inhibitory factor (LIF), which promotes JAK/STAT3 signaling and is essential to prevent differentiation (for a review, see [50]). Additionally, WNT and BMP4 signaling pathways are crucial for ESCs while fibroblast growth factor (FGF) signaling is detrimental. A more metastable naïve pluripotent state, named the ground state of pluripotency, can be achieved by using serum-free culture media containing LIF and two inhibitors of MEK/ERK and GSK3β (referred to as 2i medium), which promote WNT signaling while inhibiting the FGF pathway (for a review, see [51]). One could speculate that the 2i growth conditions may have opposite effects on pathways controlling translation. On the one hand, 2i-mediated MEK/ERK inhibition may repress MNK signaling as well as mTORC1 signaling, since MEK/ERK also stimulates mTORC1. On the other hand, as GSK3β inhibits mTORC1, 2i inhibition of GSK3β may relieve mTORC1 repression [47]. Therefore, the use of 2i medium might profoundly affect global translation in ESCs. However, further studies will be required to investigate mechanistic differences in mTOR and MNK activities in serum and 2i conditions, as well as to address whether these differences impact global or gene-specific translational efficiency and are required to actively establish the ground state of pluripotency.

#### 2.3.4. eIF2α Phosphorylation

In contrast to the mTOR and MNK pathways, stress conditions arising from amino acid starvation or ER stress can inhibit translation through the phosphorylation of the α subunit of eIF2, eIF2α. Phosphorylated eIF2α (P-eIF2α) sequesters its guanine nucleotide exchange factor eIF2B, which can only exchange GDP for GTP on unphosphorylated eIF2α As eIF2B is no longer available for recycling GDP-bound eIF2α, this causes a reduction in active 43S PIC available for initiating new rounds of translation initiation.

Friend and colleagues [12] found that eIF2α is highly phosphorylated in human and murine ESCs while differentiated ESCs display low levels of eIF2α phosphorylation. Interestingly, they also demonstrated that eIF2α phosphorylation is under the direct control of LIF and BMP4 signaling pathways, which are crucial for maintaining ESC pluripotency. Thus, while LIF alleviates the expression of the eIF2α phosphatase CReP (constitutive repressor of eIF2α phosphorylation), BMP4 promotes the activation of the eIF2α kinase PKR (protein kinase regulated by RNA), therefore synergistically promoting eIF2α phosphorylation. More interestingly, the expression of NANOG and C-MYC, the mRNAs of which contain uORFs as previously described [7], is stimulated by eIF2α phosphorylation, further suggesting that ESCs can support the translation of specific transcripts, including those harboring uORFs, in a context of global translational repression. In addition, eIF2α phosphorylation is also elevated in MuSCs and is required to maintain self-renewal and regenerative capacities [28]. Satellite cells expressing a non-phosphorylatable eIF2α mutant have an elevated protein synthesis and activate myogenic programs, indicating that they are undergoing differentiation. Similarly to ESCs, eIF2α phosphorylation controls the selective translation of uORF-containing mRNAs, such as Uspx9, which are required for MuSC self-renewal and homeostasis.

#### 2.3.5. RNA Granules

Cellular membrane-free granules play a wide variety of roles in different mechanisms and are formed by a process called liquid-liquid phase separation (for a review, see [52,53]). RNA/protein complexes termed ribonucleoproteins (RNPs) can also aggregate and form similar cytoplasmic liquid droplet structures, among which stress granules (SGs) and processing bodies (P-Bodies) are the most documented. Such structures can be constitutively present to some extent (P-bodies) or assemble in response to specific stresses (SGs) and are essential for regulating cell homeostasis in response to diverse environmental fluctuations. SGs are thought to be involved in stress-induced translational repression and are composed of translationally silent mRNAs as well as translation factors and repressors [52]. In contrast, P-bodies are thought to constitute sites of mRNA storage when decay is limiting and are composed of decay-targeted mRNAs and decay factors. Both SGs and P-bodies regulate mRNA fate by segregating them away from active translation, although this process is reversible and highly dynamic.

Interestingly, conditions that induce eIF2α phosphorylation drive the formation of SGs containing translationally repressed mRNAs [54]. Thus, RNA granules resembling SGs could play a role in stem cells containing high levels of P-eIF2α, although the presence of such structures has not been formally addressed [12,28].

Recently, Di Stefano and colleagues [55] highlighted the role of P-Bodies in ESC, iPSC, and ASC homeostasis. Indeed, the depletion of the RNA helicase DDX6, which disrupts P-bodies, inhibits ESC differentiation and causes the acquisition of a “hyper-pluripotent” state. Interestingly, DDX6-depleted primed ESCs acquired characteristics of naïve pre-implantation ESCs. P-Body integrity is essential for DDX6 effects on pluripotency, as disruption of these RNA granules by depleting other P-Body factors phenocopied DDX6 depletion. The authors demonstrated that DDX6-containing P-Bodies repress the translation of mRNAs encoding transcription factors, including OCT4 and NANOG, chromatin remodelers, and stem cell factors. In contrast with ESCs, the suppression of DDX6 has contradictory effects on ASCs. Indeed, while it reinforces self-renewal and stemness properties in NSCs and intestinal stem cells (ISCs), DDX6 depletion stimulates the differentiation of mesoderm-derived MuSCs and mesenchymal stem cells (MSCs). Thus, P-Bodies are required to repress the translation of pluripotency factors during ESC differentiation and play context-dependent roles in ASCs. Interestingly, Yang et al. [27] showed that 4E-T, which binds to eIF4E and sequesters eIF4E-bound transcripts into P-Bodies, is also required to maintain NPCs and represses translation of pro-differentiation factors, thereby preventing neurogenesis. These observations seem to contrast with those of Di Stefano et al. [55] although it should be underlined that they were drawn from different biological models, which may explain these discrepancies. However, one may speculate that specific transcripts may follow alternate P-Body assembly routes (DDX6 vs. 4E-T) and therefore may undergo alternative translational regulation. 

In conclusion, stem cells establish specific translational regulation via a global control of translation mediated by mTOR, MEK/ERK, and P-eIF2α pathways as well as through distinct translational regulation via mRNA-specific cis elements or localization. Thus, stem cells can translate specific stemness genes and repress global translation of unnecessary factors or differentiation factors while maintaining the ability to rapidly switch between translation programs in response to fate transition inputs. 

## 3. Ribosome Biogenesis in Stem Cells

### 3.1. Overview of Ribosome Biogenesis

The translation machinery is composed of a small (40S) and a large (60S) subunit, which respectively contain one rRNA (18S) and 33 proteins, and three rRNAs (28S, 5.8S, and 5S) and 47 proteins. The assembly of such large and complex molecular machinery is the most energy-consuming process in cells and requires a multi-step and highly regulated biogenesis. Ribosome biogenesis is one of the few processes to take place in several cellular compartments: After being initiated in the nucleolus, a compartment in which most maturation steps take place, it then proceeds to the nucleoplasm and is completed in the cytoplasm. It has been extensively studied in yeast as well as in higher eukaryotes, both in normal and pathological contexts, and is well documented in several recent reviews [56,57,58,59]. As previously described for translation, ribosome biogenesis is directly regulated by the extracellular environment and stresses, nutrient availability, and cell proliferation.

Briefly, the assembly and maturation of ribosomes occurs co-transcriptionally and requires over 200 factors, including proteins but also non-coding RNAs called small nucleolar RNAs (snoRNA), which are assembled with proteins to form small nucleolar particles (snoRNP). While the 5S rRNA is transcribed by the RNA polymerase III (Pol III), the 5.8S, 18S, and 28S rRNAs all arise from the processing of a common precursor rRNA (47S) transcribed by the RNA Polymerase I (Pol I). The maturation of the 47S pre-rRNA requires a series of coordinated endonucleolytic and exonucleolytic cleavages, as well as over 200 post-transcriptional base and ribose modifications to generate the mature 5.8S, 18S, and 28S rRNAs. The two most frequent modifications, 2’-*O*-ribose methylation (2’-*O*-Me) and pseudouridylation (Ψ), are respectively processed by box C/D snoRNPs together with the methyltransferase fibrillarin (FBL), and box H/ACA snoRNPs bound to the uridine isomerase dyskerin (DKC1), based on sequence complementarity between snoRNAs and rRNAs. Each pre-ribosomal subunit (pre-40S and pre-60S) is processed separately and undergoes a large number of coordinated remodeling transitions until the last maturation steps, occurring in the cytoplasm [58].

In the context of stem cells, which either need to sustain a pluripotent/multipotent “unstable” state or to coordinate gene expression to ensure specific phenotypic changes, precisely controlling the machinery of translation and its biogenesis is essential to ensure proper fate decisions. Accordingly, over the past 10 years, numerous reports established that the regulation of rRNA expression and post-transcriptional modifications plays a pivotal role in controlling stem cell fate, proliferation, and differentiation, both during development and in adult tissues. Interestingly, such regulations have been described in many stem cell models across higher eukaryotes and therefore constitute a key hallmark of stem cell physiology.

In this section, we will review evidence demonstrating that the expression of both rRNAs and ribosome biogenesis factors (RBFs) is tightly linked to the status of stem cells (Figure 2). We will also discuss reports establishing that the activity of the biogenesis machinery is regulated and coordinated with other steps of gene expression to ensure proper ribosome quantity and activity, which directly impact stem cell homeostasis or cell identity.

### 3.2. The Expression and Modifications of rRNAs Are Highly Regulated in Stem Cells

Ribosomes are composed of about 50% of RNA and are considered to be ribozymes since rRNAs carry the catalytic active site enabling peptide bond formation during protein synthesis. The transcription as well as the coordinated maturation and modification of rRNAs are therefore promising candidates to modulate and adapt ribosome synthesis to changes in stem cell identity.

Accordingly, a first landmark study established that female drosophila germ stem cells (GSCs) display higher rRNA transcription rates than their immediate differentiating progenies in ovarioles [60]. The rRNA transcription rate is controlled by a Pol I regulatory complex composed of TAF1B, TAF1C, and UDD, the latter exhibiting a high level of expression in GSCs compared to their immediate progenies. Interestingly, impairing Pol I transcriptional activity causes reduced GSC proliferation while increasing its activity induces a loss of progenies or a delayed cyst differentiation in vivo [24,60]. In addition, the depletion of pre-40S or pre-60S ribosome biogenesis factors and of a box H/ACA snoRNP responsible for rRNA pseudouridylation causes drosophila GSC differentiation [24]. Similarly, rRNA transcription rates are reduced by about 50% in a matter of hours during endodermal-lineage specification of human ESCs, as a consequence of the loss of Pol I transcription factor UBTF binding to rRNA gene promoters [61]. However, while the differential regulation of rRNA transcription and biogenesis between stem cells and their progeny seems to be a mechanism conserved in different stem cell types, the modulation of these processes appears to be model dependent. Indeed, although ESCs and GSCs express higher levels of rRNAs compared to their daughter cells, rRNA biogenesis undergoes an opposite regulation in HSCs. Mouse HSCs express significantly lower levels of pre-rRNAs as well as mature 18S and 28S rRNAs compared to committed hematopoietic progenitors, regardless of the cell cycle steps [26,62]. Interestingly, in HSCs, the rRNA expression rate is perfectly correlated with the translational activity measured in these cells, as described above. Such differences in rRNA transcription rates in different stem cell populations might either reflect their proliferative status, influences from the niche, and in vitro growth conditions, or highlight molecular mechanisms controlling rRNA expression in a cell-specific manner.

Accordingly, several other mechanisms and factors, including histone variants, chromatin regulators, and transcription factors, have recently been described to specifically control rRNA expression in stem cells. Initial observations made in drosophila GSCs and NSCs shed light on a mechanism directly connecting self-renewal and fate decisions to ribosome biogenesis. Indeed, in both ASC models, components of the Pol I transcription regulation complex (UDD) or of the U3 snoRNP complex (Wcd) were shown to segregate unevenly during asymmetric cell division and were preferentially retained in GSCs rather than in differentiating progenies [60,63]. Eleuteri and colleagues [64] elegantly demonstrated that H2A.X, a histone variant highly expressed in mouse ESCs, is abundant on rDNA promoters and actively recruits the nuclear remodeling complex (NoRC), a major repressor of rRNA transcription. NoRC silences rRNA expression and therefore lowers the ESC proliferation rate [64]. Similarly, the polycomb repressive complex 1 (PCR1)-specific protein CBX4 maintains the homeostasis of MSCs by recruiting the heterochromatin protein KRAB1 (KRAB-associated protein 1) to control the nucleolar architecture as well as to repress rDNA transcription and to prevent the accumulation of mature 18S and 28S rRNAs [65]. However, if the loss of CBX4 leads to an accelerated cellular senescence of human MSCs and is associated with MSC aging, it has no significant impact on ESC or NSC phenotypes. Finally, the RUNX1 transcription factor directly binds to repeats within rDNA promoter regions and ensures proper levels of rRNA expression and biogenesis in HSCs [66]. This gene is frequently mutated in myelodysplastic syndromes and leukemia. The loss of RUNX1 in HSCs leads to reduced rRNA levels, decreased ribosome biogenesis, and lowered protein expression. It is also associated with lower p53 levels, decreased sensitivity to apoptosis, and increased resistance to endogenous and genotoxic insults, which provide a selective growth advantage over wild-type HSCs [66]. Interestingly, the deficiency in ribosome biogenesis induced by RUNX1 depletion can be partially compensated by increased mTOR signaling, a pathway mutated in leukemia. This suggests that leukemia-causing mutations converge to maintain low ribosome biogenesis and low global translational efficiency in patient HSCs, as a general mechanism to promote increased stress resistance and long-term accumulation.

Altogether, these data clearly demonstrate that rRNA transcription is strongly regulated in stem cells, in a context-specific manner, to either sustain stem cell identity or favor their differentiation. Whether this regulation is directly sensed by factors controlling stem cell fate or indirectly affects stem cell translational capacity that in turn impacts the expression of key fate-deciding factors remains to be clarified in future studies.

### 3.3. Levels of Ribosome Biogenesis Factors Are Coordinated to Support Stem Cell Fate

Despite the importance of translational regulation for the control of cell identity as previously discussed, the global expression landscape of RBFs and the regulation of their activity remain poorly documented in the context of stem cells. Regulation of ribosome biogenesis may be achieved by alternative means, by either globally regulating RBF expression or precisely modulating the stoichiometry of specific RBFs. In this section, we will review reports supporting both hypotheses (summarized in Figure 2).

#### 3.3.1. Global Regulation of RBF Expression During Changes in Stem Cell Identity

Although not specifically addressed to date, several large-scale analyses of gene expression regulation between stem cell models and their differentiated progenies, in different tissues or developmental contexts, support the hypothesis that the ribosome biogenesis machinery undergoes coordinated changes as cell identity undergoes reprogramming. The first lines of evidence arise from a proteomic analysis of the RNA binding protein (RBP) landscape expressed in mouse ESCs, which revealed that 13 RBFs, including NMD3, NIP7, NOP56, NOP58, and RPF1, are expressed at higher levels in pluripotent cells compared to their differentiated progenies [67]. 

During hematopoiesis, a large number of pre-ribosome-associated RBFs and RPs have been shown to be co-expressed at higher levels in human HSCs and in common myeloid progenitors or granulocyte progenitors compared to differentiated monocytes [68]. This observation is also corroborated by a single-cell analysis of zebrafish hematopoietic lineages for many RBFs, including Nucleostemin (NS), Fbl, Nop58, Nop16, Nop10, Nucleolin (Ncl), Drg1, and Nop56, suggesting that upregulation of genes encoding RBFs in ASCs may be a conserved mechanism [69]. In contrast to these results, a single cell analysis of mouse brains covering 25 different neural and glial lineages revealed that components of the ribosome biogenesis machinery display a highly variable expression in NSCs, neural, and glial progenitors, as well as in differentiated cells [70]. In particular, some components are upregulated in undifferentiated cells (e.g., FBL, NCL, NOP56, NOP58, PHAX, and RSL24D1) while others are more expressed in differentiated progenies (NIP7, GAR1) or are ubiquitously expressed in all cell lineages. Surprisingly, the comparison of the same cell populations from young and aged brains revealed that young NSCs express more RP-encoding mRNAs compared to aged NSCs while aging seems to globally promote an increase in RP expression across differentiated cells. Similarly, the peripheral midbrain layer of zebrafish embryo is enriched in slow amplifying progenitor (SAP) cells, which specifically express high levels of nucleolar RBFs, including Nop56, Nop58, Fbl, Npm1, Nle1, Wdr12, Wdr46, and Pes1. These proteins are considered to be markers of SAP cells [71]. Altogether, these reports suggest that expression levels of RBFs are highly regulated and globally enriched in stem cells, yet specific factors show a more restricted regulation in differentiated lineages, which could reflect peculiar functional or proliferative requirements of more mature cell states.

#### 3.3.2. Specific Ribosome Biogenesis Factors Are Preferentially Expressed in Stem Cells

In addition to the large-scale analyses detailed above, an increasing number of studies have established that individual RBFs are specifically enriched or expressed in stem cells. Firstly, the expression of PHAX (phosphorylated adaptor for RNA export), an RBF involved in intranuclear transport of snoRNPs, is regulated during the commitment of human ESCs towards hematopoietic lineages [72]. Similarly, the E3 ubiquitin ligase UBR5 is highly expressed in human iPSCs and is downregulated during neural differentiation [73]. In addition, FBL and NCL are significantly downregulated as murine ESCs differentiate in the absence of LIF or form EBs, respectively [74,75]. The loss of expression of these proteins in ESCs leads to cell growth arrest, activation of the p53-dependent pathway, and ESC differentiation. Similarly, DKC1, which catalyzes rRNA pseudouridylation, and the zinc finger nucleolar protein LYAR, are expressed at higher levels in mouse ESCs compared to differentiated progenies [76,77]. Interestingly, LYAR has been shown to interact with and to stabilize NCL, thereby sustaining a stable NCL steady-state expression in ESCs. Similarly to NCL and FBL depletion, the downregulation of LYAR in mouse ESCs causes proliferation arrest and increases apoptosis, which are likely partially explained by the loss of NCL stability. However, LYAR depletion also impairs ESC differentiation as levels of key pluripotency factors, such as OCT4 and NANOG, remain elevated [77]. Pluripotent cells therefore seem to have specific requirements for rRNA modification enzymes compared to their differentiated progenies.

In postnatal tissues, the expression of notchless (NLE), a biogenesis factor involved in the maturation of the pre-60S subunit, is enriched in murine HSCs compared to different immature progenitors and bone marrow cells [78]. NLE is also highly expressed in mouse intestinal stem cells and progenitor cells [79] and its expression is increased upon activation of quiescent MuSCs [80]. Similarly, the nucleolar protein NS, a putative RBF [81,82,83], is expressed in murine pre-implantation embryos, in ESCs, MSCs, NSCs, HSCs, and hematopoietic progenitor cells as well as in spermatogonia and is rapidly lost upon differentiation [84,85,86,87,88]. Along this line, NS is a marker of GSCs during prepubertal spermatogenesis [85] and of different types of cancer stem cells [89]. Altogether, these reports demonstrate that even in absence of evidence supporting a global ribosome biogenesis machinery upregulation in stem cells, the expression regulation of individual key RBFs is required for the maintenance of stem cell properties both in ASCs and ESCs.

### 3.4. Ribosome Biogenesis Is Highly Regulated to Support Stem Cell Properties

In addition to evidence supporting correlation between RBF expression and stemness or differentiated identities, in a developmental context or in adult tissues, several studies have described the functional implications of proteins involved in ribosome biogenesis in stem cell homeostasis, regardless of their expression profiles. The importance of these factors in maintaining either stem cell identity or proliferation and their molecular functions in ribosome maturation will now be discussed.

#### 3.4.1. Pre-Ribosomal Subunit Maturation Control in Stem Cells

A first line of evidence emerges from studies in ASCs. Using an RNAi screen approach in adult drosophila ovaries, Sanchez and colleagues [24] established that different components of the eIF3 complex and of the box H/ACA snoRNPs (NOP10, GAR1, DKC1, and NHP2) are required for GSC maintenance in vivo. The depletion of these factors either induces a loss of GSCs or impacts early or late differentiation events of germline development, thereby highlighting a pleiotropic impact of alterations in ribosome biogenesis on GSC fate.

Similarly, in drosophila testis, the small ribonucleoprotein particle protein SmD3 was reported to play an essential role in maintaining GSC homeostasis and testis functions by interacting with and controlling the expression of spliceosome components and of a large panel of RPs, including RPL18 [90]. You and colleagues [16] conducted a large siRNA screen against RBPs and identified nine RBFs required for ESC maintenance and NANOG steady-state expression [16]. Interestingly, they identified that six of these factors belong to the pre-18S rRNA processing complex (SSUP), including KRR1, WDR46, and MPP10, which are downregulated upon ESC differentiation in EBs. Depletion of KRR1 caused a downregulation of many key pluripotency genes and significantly reduced the reprogramming efficiency of iPSCs. At the molecular level, the depletion of KRR1 caused an accumulation of 30S pre-rRNA intermediates, a loss of the 40S subunit and 80S ribosomes, and a decrease in global translation. The authors concluded that despite a global reduction in protein synthesis, the loss of pluripotency was likely the result of a rapid decrease in unstable pluripotency transcription factors, including NANOG and ESSRB, which are critical for ESC maintenance. Similarly, Bennett and colleagues [91] established that the DEAD-BOX RNA helicase DDX27 is highly expressed in Pax7-positive MuSCs in zebrafish and mouse, and decreases upon myofiber differentiation [91]. DDX27 depletion perturbed the nucleolar architecture and caused an accumulation of early pre-rRNA precursors concomitant with a reduction in the accumulation of mature rRNAs. DDX27 loss consequently impaired the association of polysomes with a subset of mRNAs, including transcripts involved in protein biosynthesis and muscle growth. These molecular defects in satellite cell differentiation cause a premature activation of myogenic programs, muscle hypotrophy, and reduced muscle contractility.

In two other compelling studies, Cohen-Tannoudji and colleagues [78,80] established that NLE, a pre-60S maturation factor, is required for the maintenance of mouse HSCs and the proliferation of muscle satellite cells. In contrast to its enriched expression in HSCs and immature hematopoietic progenitors, NLE was expressed at low levels in quiescent satellite cells and significantly increased in activated MuSCs, perfectly matching rRNA expression in this model. 

Surprisingly, while the depletion of NLE in HSCs causes a rapid exhaustion and loss of both HSCs and immature progenitors, NLE inactivation in activated MuSCs caused a proliferation arrest and reduced levels of phosphorylated RPS6. However, in both models, NLE expression seems to be dispensable for the proliferation of more restricted progenitors as well as for the maturation of differentiated myofibers and quiescent satellite cells. At the molecular level, NLE inactivation induced an accumulation of pre-28S rRNA intermediates associated with an activation of the p53 pathway in both stem cell models, and a significant decrease in the 60S subunit and 80S ribosomes in HSCs. These data therefore demonstrate that stem cell-specific requirements in ribosome biogenesis can be modulated by a single RBF as a mean to allow a rapid adaptation of ribosome production in response to changes in fate.

#### 3.4.2. Regulation of rRNA Modifications

In addition to factors impacting general aspects of ribosome biogenesis, several reports have established that the molecular mechanisms controlling rRNA post-transcriptional modifications, in particular the methylation of the ribose 2’ hydroxyl group (2’*O*-Me) and the pseudouridylation (Ψ), are highly regulated in stem cells, as previously suggested. Firstly, the ribosome biogenesis proteins UBR2 and UBR5 are expressed at high levels in zebrafish HSCs, as well as in human and murine ESCs, respectively [92,93]. UBR5 interacts with box H/ACA snoRNPs and its depletion impairs the production of mature ribosomes in mouse ESCs. Although the molecular impact of UBR2 on ribosome biogenesis has not been addressed, both UBR2 and UBR5 depletion cause a p53 pathway-dependent reduction in HSC and ESC proliferation. Similarly, the nucleolar protein LRRC34 is enriched in mouse ESCs, interacts with two important RBFs, NCL and NPM1, and is required for pluripotent cell proliferation [94]. 

A landmark paper recently and convincingly established that NPM1 is a key mediator of rRNA post-transcriptional modifications in mouse embryonic fibroblasts (MEFs) and HSCs [95]. Nachmani and colleagues [95] demonstrated that NPM1 is associated with a large number of box H/ACA and box C/D snoRNAs, and that the 2’*O*-Me levels of five 28S rRNA sites are specifically and significantly impaired in Npm1^−/−^ MEFs, yet no changes in Ψ were detected. NPM1 directly interacts with FBL to promote the correct assembly of box C/D snoRNAs into snoRNPs. Interestingly, two of the NPM1-dependent 2’*O*-Me are localized in the peptidyl transferase center. Although the general cap-dependent translation remained unaffected, NPM1 depletion impaired IRES-dependent translation in a snoRNA-dependent manner. Finally, NPM1 mutant mice present a highly penetrant altered hematopoiesis, similarly to phenotypes observed in ribosomopathies, together with HSC hyperproliferation and decreased differentiation capacities, which overall recapitulate the AML phenotypes of patients harboring NPM1 mutations. NPM1 and DKC1 were also shown to form complexes with the pluripotency factors OCT4, NANOG in ESCs [76,96]. DKC1 is described as an important transcriptional co-activator of OCT4/SOX2 complexes and is required for both ESC maintenance and iPSC reprogramming. However, the functional roles of these interactions and of NMP1- and DKC1-dependent rRNA modifications, namely 2’*O*-Me and Ψ, in pluripotent stem cells remain elusive [96].

Finally, FMRP (Fragile X Mental Retardation Protein), the loss of which is responsible for Fragile X syndromes, was shown to alter adult murine NSC differentiation capacities [97], mouse cognitive capacities [98], and human ESC neuronal differentiation [99]. Expression and ribosome profiling assays in murine wild type and *Fmr1* KO NSCs revealed that loss of FRMP impairs both mRNA expression and translation of several neurogenesis and synaptic genes, and reduces the expression of 17 mitochondrial RP genes [100]. Interestingly, the drosophila FRMP represses translation by directly interacting with the 60S ribosomal subunit through RPL5 [101]. More recently, FMRP was shown to interact with several box C/D snoRNAs in human ESCs and NPCs, and to mediate differential 2’*O*-Me of 12 partially methylated sites of the 18S and 28S rRNAs while constitutively methylated sites remained invariable [102]. FMRP plays an essential role in stem cells by coordinating both rRNA and RP expression to control fate decisions. Furthermore, FMRP binds to ribosomes carrying specific methylation patterns on three variable sites, suggesting that FMRP may contribute to generating ribosome heterogeneity at the molecular level, a concept that will be discuss in the fourth section of this review.

The functional importance of rRNA modifications on ribosome maturation and activity remain to be extensively studied in the context of stem cells. Indeed, with the exception of FMRP previously discussed, and despite several studies demonstrating the importance of FBL, DKC1, or snoRNP factors in stem cells, the impact of these factors on rRNA 2’*O*-Me or Ψ remains largely unknown.

#### 3.4.3. Ribosome Biogenesis Surveillance by P53-Dependent Pathways in Stem Cells

p53 is a well-established tumor suppressor gene and an essential sensor of cellular stresses. Briefly, in mouse normal cells, p53 activity is limited by its interaction with the E3 ubiquitin ligase MDM2 (also known as HDM2 in human), which constitutively promotes p53 ubiquitination and degradation by the proteasome. In the context of ribosomal stress, NPM1 or free RPs from destabilized ribosomal subunits can directly interact with MDM2/HDM2, thereby disrupting the p53 interaction and promoting p53 protein stabilization as well as the activation of p53 downstream target genes driving an apoptotic response (for detailed reviews, see [103,104,105]). Interestingly, activation of the p53-mediated surveillance of ribosome biogenesis has been proposed to contribute to the pathophysiology of a group of diseases called ribosomopathies, which are caused by mutation of RBFs and RPs and are characterized, among others, by a higher susceptibility to develop cancer (see these excellent reviews for details on ribosomopathies [106,107,108,109,110]). As previously described, alterations of nucleolar protein and RBF expression implicated in pre-ribosome subunit maturation (NCL and NLE) or rRNA modifications (UBR2 and UBR5) cause ribosome biogenesis defects and result in p53-dependent cell cycle arrest or apoptosis in different models of ASCs and ESCs [75,78,80,92,93]. Similarly, several reports have established that haploinsufficiencies of distinct RPs also trigger ribosomal stress and activate the p53-dependent apoptotic pathway in mouse ESCs, iPSCs [111], and human erythroid progenitor cells [112], leading to a loss of stem cell maintenance. Similarly, Fortier and colleagues [113] established that haploinsufficiency of several RPs differentially affects the capacity of ESCs to differentiate into EBs: e.g., ESCs hemizygous for RPS5 show an impaired expression of mesoderm-specific genes. Interestingly, apoptosis induced by the ribosomal stress pathway can be rescued by p53 inactivation [112,113] or downregulation [78,80], restoring a near-to-normal stem cell homeostasis. However, decreased levels of several RPs, in particular RPS5, RPS14, and RPS28 proteins located at the mRNA exit site in ribosomes, seem to trigger p53-independent ribosomal stress as p53 inactivation does not rescue stem cell survival [113]. Altogether, these data suggest that, similarly to somatic cells, the ribosomal stress pathway acts as a surveillance pathway in stem cells, likely to eliminate cells carrying mutations or mis-expressing ribosome components that affect ribosome biogenesis and could impair stem cell fate.

### 3.5. Coordination Between Ribosome Biogenesis and Additional Steps of Gene Expression

The maintenance of stem cell identity as well as the induction of changes in fate during differentiation require a complex regulation of genetic programs, which is achieved by a perfect coordination of several gene expression processes, such as chromatin, epigenetic, transcriptional, and translational control. Interestingly, converging evidence published in the past two years also establishes that ribosome biogenesis is also tightly coordinated to precisely adapt the ribosome content to cellular status and demand. The first line of evidence arises from multitask RBFs, such as NS, NCL, NPM1, FMRP, and UBR5, which are implicated in DNA damage response, transcription, RNA shuttling, and protein degradation [75,89,114,115,116]. Moreover, as previously discussed, rRNA expression is tightly coordinated with chromatin structure (H2AX) [64], epigenetic modifications [65], and transcription (e.g., RUNX1) [66] in stem cells. Ribosome biogenesis also seems to be highly coordinated with the maturation of the RNA splicing machinery by SmD3 in drosophila GSCs [90]. Finally, Corsini et al. [117] established that the RBF HTATSF1 interacts with multiple splicing factors, including SF3B1, as well as rDNA transcription regulators in ESCs. More precisely, HTATSF1 coordinates rRNA expression and protein synthesis. Such HTATSF1-dependent protein synthesis was shown to be important for both in vitro ESC fate regulation and epiblast development. 

In addition to RBFs, the expression of RPs is tightly coordinated in NSCs [70], HSCs [68], and ESCs [7,20,67,117]. Indeed, HTATSF1 has first been shown to control intron retention of multiple RP pre-mRNAs, thus controlling the expression of the corresponding proteins in ESCs and during early embryogenesis [117]. HTATSF1 therefore acts as a master regulator of multiple post-transcriptional mechanisms to coordinate the level of protein synthesis required to maintain pluripotency in ESCs. Interestingly, RP mRNAs also possess a conserved feature called 5‘-terminal oligopyrimidine tract (TOP), which starts with a C at the first position next to the 5’ cap and is followed by an uninterrupted stretch of 4-14 pyrimidine residues [118,119]. mRNAs carrying a TOP sequence are strongly regulated at the translation level and follow a binary mode of translational control that have been termed the “all-or-none” model (review in [120]). Although the signaling pathways controlling translation of TOP mRNAs are still under intensive investigations, it seems that RPS6 phosphorylation by S6Ks, mTOR activity, and the PI3-kinase pathway activation in response to growth or mitotic signals is essential for the translational regulation of TOP mRNAs (for review [118,121]). Interestingly, in mouse ESCs, the translation of TOP mRNAs is regulated in response to differentiation signals by the PI3-kinase pathway independently of RPS6 phosphorylation [122]. In addition, as discussed in Section 2, the mTOR pathway and S6Ks activity are finely regulated in ESCs [10,11,48], iPSCs [15,49], and ASCs [22,24,26], suggesting that the control of RP expression in stem cells and during cell fate transitions [7,20,67,117] might partially be mediated by TOP regulatory sequences. 

Altogether, these data demonstrate that the expression of ribosome components (RBFs and RPs) is tightly coordinated by different mechanisms, including epigenetics, transcription, post-transcriptional maturation mechanisms, translation, and signaling pathways. This regulation therefore emerges as a mechanism to control stem cell homeostasis and identity changes.

## 4. The Ribosome, a New Regulator of Translation in Stem Cells

As previously discussed, both conventional mechanisms of translational regulation and ribosome biogenesis actively contribute to germ, somatic, and embryonic stem cell homeostasis, either by maintaining their specific identity or by actively contributing to genetic rewiring during cell fate transition. Until recently, ribosome subunits (40S and 60S), monosomes (80S), and polysomes were considered to have a perfectly stoichiometric RP composition as well as an inflexible rRNA content and a well-defined repertoire of post-transcriptional modifications among all cells and tissues. Nevertheless, it has been known for a long time that RP expression as well as snoRNA expression, which guide rRNA modifications, are highly variable between tissues and cell types, although the impact of these fluctuations on ribosome composition remains unclear. In addition, whether putative heterogeneous ribosome populations may acquire specific functions is intensively debated.

Recent observations that ribosome production is highly regulated in stem cells and participates in cell identity and changes in fate, together with long-standing observations described above, raised several intriguing questions: Does stem cell ribosome biogenesis impact ribosome abundance, composition or both? How do these quantitative or qualitative changes in ribosome content impact stem cell-specific gene expression? Is the role of ribosomes in gene-specific regulations conserved in other cell types and tissues? In this last part, we will discuss current hypotheses on how ribosomes, previously considered to be devoid of regulatory activities, may specifically regulate gene expression (Figure 3). We will then review evidence supporting a presumed molecular plasticity of ribosomes between tissues and discuss recent observations supporting the presence of functionally specialized ribosomes in stem cells.

### 4.1. The Ribosome Concentration Regulates the Expression of Specific Transcripts

The contribution of ribosomes to gene-specific regulations of translation was largely understudied until recently, when it gained attention following the extensive characterization of a group of pathologies called ribosomopathies. This group of diseases is phenotypically highly heterogeneous, although common defects in ribosome production caused by mutations in RBF or RP genes occur [123,124]. Considering the importance of ribosome in cell biology as a crucial and constitutive molecular machinery, it is quite puzzling that only some organs are affected while others can tolerate and compensate ribosome dysfunctions. Hence, intensive studies investigated how ribosome deficiencies can cause tissue-specific defects, and several hypotheses discussed in excellent reviews have been proposed to explain these observations [106,107,108,109,110]. While some hypotheses suggest that the activation of p53-dependent ribosomal stress upon RP deficiency (described in Section 3) may explain organ-specific dysfunctions in ribosomopathies, others suggest that phenotypes may be supported by models implying that ribosomes possess translational regulatory activities and mRNA selectivity in specific cell types. For the purpose of this review, we will focus on models that propose a contribution of ribosomes to the regulation of translation and discuss observations supporting such a model in stem cells.

First, based on mathematical models and observations, Mills and Green proposed that the “ribosome concentration” could affect the translation of specific transcripts while only moderately affecting others (Figure 3) [108]. Indeed, they hypothesized that only peculiar cell types, which rely on the expression of specific transcripts, could be affected by perturbations of ribosome concentrations, while global gene expression might be moderately impacted in other cell types. This model reasonably assumes that differences in the translation initiation rate render mRNAs variably dependent on cellular ribosome abundance. Accordingly, when the ribosome concentration is low, mRNAs with high or moderate translation initiation rates are properly translated, while mRNAs with low translation rates are not (Figure 3, upper panel). However, when the ribosome concentration is high, mRNAs with low or moderate translation initiation rates are properly translated while translation of mRNAs with very efficient initiation may be dampened or abrogated via a “ribosome crowding” effect (Figure 3, lower panel). Accordingly, mRNAs can be selectively translated upon ribosomal content fluctuations, which can be either physiological or pathological (ribosomopathies). Importantly, as previously mentioned, mRNAs carrying IRESs, uORFs, or complex 5’UTR structures, which encode for key stem cell factors and are usually inefficiently translated, may be particularly sensitive to ribosome concentration fluctuations. 

Hence, in the context of ribosomal concentration deficiencies caused by ribosomopathies, mRNAs with low translation initiation rates will be more impacted than mRNAs translated with higher initiation rates. This hypothesis is supported by the translation impairment of GATA1 mRNAs, an important erythroid transcription factor in the context of Diamond–Blackfan anemia (DBA), which is characterized by a profound anemia associated with developmental defects, such as skeletal abnormalities [125]. Interestingly, GATA1 mRNAs harbor a highly structured 5’ UTR that diminishes the GATA1 translation initiation rate. In the absence of RP haploinsufficiency, GATA1 mRNA translation is sufficient to support bone marrow cell lineage production (Figure 3) while DBA-induced ribosome deficiencies particularly affect GATA1 translation initiation and expression. Accordingly, the erythropoiesis-deficient phenotype associated with DBA can be overcome by artificially increasing GATA1 protein levels [125]. This perfectly illustrates that changes in the ribosome concentration may affect gene-specific expression in a pathological context.

While the “ribosome concentration” model has been proposed to explain tissue-specific phenotypes in the context of ribosomopathies, direct evidence that regulation of the ribosome concentration controls stem cell-specific gene expression and participates in cell identity transitions is lacking. However, some indirect observations support that stem cells are particularly sensitive to changes in ribosome concentration. First, there is evidence that RP expression varies while ESCs undergo differentiation, although whether these changes affect ribosome concentration or simply the free pool of RPs has not been rigorously addressed (Section 3 [7,20]). In addition, Fortier and colleagues [113] demonstrated that deficiencies in RP expression affect ESC differentiation and that a part of these effects are independent of the p53-mediated nucleolar stress response. Therefore, one could speculate that either RPs possess extraribosomal functions that control ESC differentiation or RP deficiencies affect ribosome concentration and thereby the expression of differentiation programs. However, this later hypothesis is highly speculative and will require more thorough demonstrations.

Although physiological fluctuations of ribosome abundance remain poorly understood, translation and ribosome concentration were shown to be regulated by the circadian rhythm in plants [126] and mammals [127,128,129]. Indeed, almost 90% of mRNAs subject to circadian translation are TOP mRNAs (121), and accordingly the circadian clock controls the expression of RPs and ribosome biogenesis in the liver [127,129,130]. Moreover, FRMP regulates circadian clock mRNAs in mouse CA1 neurons [128] and the circadian clock is maintained in several models of stem cells, in particular in ASCs [131,132,133]. Altogether, these data strongly suggest that, at least in some models of stem cells, intracellular ribosome concentrations might be regulated by the circadian clock, although this hypothesis remains to be demonstrated experimentally in these specific cell types. Thus, it would be interesting to address whether the fluctuation in RP expression during circadian rhythm influences the stem cell ribosome concentration and therefore impact stem cell homeostasis.

Noteworthy, the ribosome concentration is sustained by a subtle dynamic equilibrium between ribosome production, recycling, and degradation. Thus, perturbations of ribosome recycling caused by ribosome stalling likely affect the pool of ribosomes available for new rounds of translation. Thus, rescue factors, such as PELOTA, which specifically resolve stalled ribosomes, regulate ribosome recycling and concentration. [134,135]. Interestingly, PELOTA is essential for mouse and drosophila GSC maintenance and spermatogenesis [136,137,138]. While dispensable for mouse ESC self-renewal, PELOTA is required for both extraembryonic endoderm differentiation and the reprogramming of iPSCs [139]. Despite the fact that the consequences, at the molecular level, of PELOTA loss-of-function in these stem cell models has not been demonstrated as being strictly dependent on its role in ribosome recycling, one can hypothesize that ribosome recycling pathways may also participate in ribosome concentration regulation as a means of modulating stem cell fate. 

In conclusion, whether the model proposed by Mills and Green [108] applies to physiological regulation of gene expression remains an open question. Indeed, future studies should address whether changes in ribosome biogenesis observed during stem cell fate transition impact gene-specific expression exclusively through modulation of ribosome abundance. 

### 4.2. Specialized Ribosomes as Regulators of Gene Expression?

As an alternative to Mills and Green’s proposal, tissue specificity observed in ribosomopathies could be explained by a model in which RP and RBF functions may not be ubiquitous but conversely tissue specific. Thus, specific RPs or rRNA modifications may be essential for supporting ribosome functions in particular cell types but not in others. In other words, this model implies that some ribosome populations may have molecular compositions and translational activities that differ from canonical ribosomes and may therefore acquire “specialized” cell- or tissue-specific functions. In addition to ribosomopathies, long-standing observations have supported the “specialized” ribosome model, without undoubtedly proving their existence. In this part, we will discuss observations supporting ribosome composition heterogeneity and evidence proving ribosome functional specializations and its impact on stem cell homeostasis.

#### 4.2.1. RP Tissue-Specific Expression and Functions

Bortoluzzi and colleagues [140] observed that RP mRNA expression varies greatly among tissues and cell types in mammals. Indeed, using available human expression datasets for uterus, ovary, brain, liver, skeletal muscle, and retina, they identified 13 differentially expressed RP genes, including RPL14, RPL38, RPS9, RPS10 and RPS19 (mostly expressed in ovaries), RPL37, RPL37a, RPL41, RPS17, RPS25 RPS23, RPL44, and RPS13 (mostly in skeletal muscle). Similarly, Guimaraes and Zavolan [141] reported a significant heterogeneity of RP expression in a large set of human tissues, primary cells and tumors. This heterogeneity is particularly prevalent during hematopoiesis, as primary hematopoietic lineages can be clustered strictly based on the RP expression landscape [141]. Therefore, RP expression demonstrates an unexpected degree of heterogeneity in mammalian tissues.

In addition, multiple developmentally regulated RP paralogs have been identified in plants, which may suggest a potential ribosome composition heterogeneity although there is no direct evidence that plant RP paralogs can assemble into functional ribosomes [142]. Similarly, in *S. Cerevisiae*, many RPs possess paralogs with additional non-redundant functions [143] while in contrast, most mammalian RPs are encoded by a unique gene. Although several functional paralogs have been identified in mammals, they are generally considered to be non-expressed pseudogenes [144]. In mice, 4 out of 79 RP-coding genes are located on the X chromosome: *Rps4*, *Rpl10*, *Rpl36a*, and *Rpl39* [145]. Interestingly, paralogs of *Rps4*, *Rpl10*, and *Rpl39*, respectively called *Rps4l*, *Rpl10l*, and *Rpl39l*, are exclusively expressed in the testis or ESCs [145,146,147,148,149]. Although, their respective roles remain to be clearly established, Jiang and colleagues [146] suggested that the expression of the *Rpl10l* gene, localized on an autosomal chromosome, compensates for the transcriptional silencing of the X-linked *Rpl10* gene mediated by inactivation of the meiotic sex chromosome. Hence in this context, RPL10L-ribosomes are proposed to possess redundant functions with RPL10-ribosomes, although this study has not investigated potential RPL10L-specific functions. Further studies should similarly address the role of RPS4XL- and RPL39L-containing ribosomes.

The variation of RP expression has also been observed during embryonic development. Kondrashov and colleagues [150] studied the expression of RPs in mouse embryos and revealed an unexpected heterogeneity of expression among embryonic tissues. Interestingly, RPL38 expression is enriched in the developing eye, face, neural tube as well as in somites, and RPL38 loss-of-function accordingly mostly affected these organs. However, RPL38 loss-of-function did not significantly affect global protein synthesis but altered the translation of a specific subset of mRNAs, including transcripts encoding homeobox (Hox) family members. Hox genes are involved in axial skeletal patterning, and loss-of-functions of different Hox genes phenocopied tissue-specific defects observed in RPL38 knockout mice, suggesting that RPL38 developmental defects are mostly caused by a deficiency in Hox mRNA translation. Although we cannot exclude that RPL38 extra-ribosomal functions may also impact the translation of specific mRNA subsets, one could alternatively hypothesize that the decrease of ribosome concentration upon RPL38 deletion affects the translation of specific mRNA subsets (Mills and Green Model), or that RPL38-containing ribosomes may have acquired additional functions. 

Although, the expression of RPs and of their paralogs is variable, these observations do not demonstrate the reality of specialized ribosomes in mammalian cells. Indeed, variations of RP steady-state expression may not necessarily impact ribosome composition and most studies lack systematic quantitative analyses of ribosome composition. In addition, RPs and their paralogs may have acquired additional extra-ribosomal functions that could directly or indirectly impact translation. For instance, RPL22 and its paralog RPL22L1 play distinct roles in hematopoiesis and do not have redundant functions [151]. Indeed, while RPL22 is important for late differentiation events, RPL22L1 is required for the maintenance of HSCs, and they both have antagonistic extra-ribosomal activities on Smad1 translation [151] as well as on Smad2 mRNA splicing [152].

#### 4.2.2. rRNA Modifications Contribute to Ribosome Heterogeneity

rRNAs are the second most frequently modified RNA molecules, with about 2% of nucleotides subjected to post-transcriptional modifications. 2’*O*-Me and Ψ are by far the two most common modifications of rRNAs with over 100 sites for each in mammals. Recent technological breakthroughs, such as RiboMet-seq, 2*O*Me-seq, and Ψ-seq, have prompted studies on the global landscape of 2’*O*-Me and Ψ, identified new modified sites, and quantitatively measured modification levels [153,154,155]. Interestingly, these modifications largely cluster in functionally important regions of the ribosome, including the peptidyl-transferase and decoding centers (P and A sites, respectively) and the RNA interface between the 40S and 60S subunits, indicating that some variations in modification levels likely impact translational activity.

Consistently, recent evidence established that specific 2’*O*-Me sites, which are impacted by FBL depletion in cancer cells and are localized in functional domains of the ribosome, affected the translational efficiency of IRES-containing mRNAs [156,157]. As previously mentioned in the context of ESCs and ASCs, several RBFs implicated in rRNA modifications are differentially expressed between stem cells and their differentiated progenies, suggesting that rRNAs may be differentially methylated either globally or on specific sites depending on the stem cell status. Accordingly, FMRP depletion in human ESCs impacted the methylation of 24 sites while others remained unaffected [102]. Similarly, FBL knockdown in mouse ESCs impaired 2’*O*-Me levels of only 11 sites [153]. However, in both studies, the functional impact of these variations on ribosome functions and activity has not been investigated.

Very recently, using RiboMeth-seq profiling of rRNA 2’*O*-Me sites in rodent adults and embryos (E16.5), Hebras and colleagues [158] described that the vast majority of methylated sites are close to fully methylated in adult tissues while developing tissues demonstrate distinct hypomethylation patterns and a strong dynamic of 2’*O*-Me. Interestingly, they observed that the expression of some snoRNAs, including SNORD78, is highly dynamic during development and is correlated with the methylation levels of specific sites. Additional evidence establishes that the expression of snoRNAs can be tissue specific, further supporting the possibility that specific rRNA modifications may be regulated and could generate heterogeneously modified rRNAs in physiological contexts [159]. Finally, using the 2’*O*Me-seq approach, Incarnato and colleagues [153] provided the first global mapping of rRNA 2’*O*-Me in a stem cell model (mouse ESCs). Interestingly, they revealed that 2’*O*-Me patterns are generally conserved between Hela cells and ESCs, although some differences in 2’*O*-Me levels were identified at specific sites. However, the functional impact of these site-specific variations has not yet been investigated, and whether the global 2’*O*-ME landscape is modulated upon ESC differentiation remains to be established.

Altogether, observations that rRNA is heterogeneously modified under specific conditions or in specific tissues suggest that ribosomes may show some degree of plasticity at the rRNA level. However, further studies should address whether rRNA modification fluctuations occur on bona fide mature and active ribosomes and impact ribosome functions or activity. 

#### 4.2.3. ESC Ribosomes Are Heterogeneous and Functionally Specialized

Demonstrating the existence of “specialized” ribosomes is technically complex as it requires laborious biochemical purifications and characterizations of active ribosomes, and proofs that changes in composition specifically impact ribosome functions. In addition, many studies relied on artificial RPs or on the depletion of paralogs since physiological modulations of ribosome composition are difficult to identify. *S. cerevisiae* has been an interesting model to address this question in eukaryotes. Recently, Ferretti and colleagues [160] provided a step forward in this tedious effort by convincingly demonstrating the presence of specialized ribosomes in yeast. Indeed, by initially depleting Rps26, they identified specific mRNA substrates bound by either Rps26-depleted or Rps26-containing ribosomes. Hence, they demonstrated that Rps26-containing ribosomes translate mRNAs harboring a strong Kozak consensus sequence while Rps26-deficient ribosomes bound poorly translated mRNAs encoding stress-response factors. More importantly, stress conditions induced a physiological production of Rps26-deficient ribosomes that selectively synthesize stress-response factors, therefore illustrating a physiological production of specialized ribosomes that regulate specific genetic programs.

Recent developments of sensitive and quantitative proteomic analyses as well as high-throughput methods to measure whole transcriptome translation, such as Ribo-seq, have significantly accelerated the identification of specialized ribosomes in mammals, and more precisely in ESCs. Hence, Slavov and colleagues [161] elegantly demonstrated that the RP stoichiometry differs between monosomes and actively translating ribosomes (polysomes) in mouse ESCs, suggesting that heterogeneous ribosomes may co-exist in ESCs. Interestingly, they also showed that ribosome protein stoichiometry changes in yeast between monosomes and polysomes but also upon modifications of culture conditions, suggesting a conserved feature of ribosome plasticity during evolution. Although this quantitative measurement of RP stoichiometry does establish ribosome composition heterogeneity and plasticity, it does not address whether these heterogeneous ribosome populations possess functional specializations.

In a very elegant study, the team of Maria Barna [162] recently provided a convincing demonstration of the existence of functionally divergent ribosomes in ESCs. Shi and colleagues [162] performed absolute mass spectrometry quantifications of polysomal ribosomes and identified several RPs unexpectedly present in substoichiometric amounts, including RPL10A and RPS25, demonstrating ribosome heterogeneity within ESCs. In addition, they demonstrated that RPL10A- and RPS25-containing ribosomes specifically translate distinct pools of mRNAs compared to the total ribosome population or ribosomes that contain RPL22 at stoichiometric levels. Therefore, ribosomes with distinct compositions can translate specific mRNA pools, demonstrating a functional specialization of ribosome populations in ESCs in terms of genetic expression. Interestingly, RPL10A- and RPS25-containing ribosomes seem to translate mRNA subsets with specific features, such as IRESs, and distinct biological functions, including metabolism and cell cycle. Further phenotypic analyses should address whether these heterogeneous ribosomes regulate ESC homeostasis and/or participate in cell fate transitions.

Many of the observations supporting ribosome plasticity have been made individually, either at the protein or rRNA levels. Of note, it is still unclear whether the heterogeneity of rRNA modifications, especially 2’*O*-Me, can provide ESCs with specific ribosome functions, either directly by influencing rRNA properties (some rRNA modifications are crucial for ribosome functions), or indirectly by modifying ribosome biogenesis and RP composition. Further studies should map rRNA modifications, in particular Ψ, in ESCs undergoing changes in cell identity and determine which rRNA modification sites provide ribosomes with specific properties important for cell identity and fate transitions.

Predicting translational regulation by the ribosome will therefore require additional efforts in order to combine both influences of ribosome concentration modulations (“concentration model”), which has only been studied at the scale of the tissue so far, and of ribosome composition (rRNA and RPs). Depending on the cellular model studied, one model might prevail over the other, or they might be tightly coordinated to support gene expression programs that ensure stem cell identity or changes in fate.

## 5. Conclusions

The intrinsic properties of stem cells, which enable them to maintain a specific identity while remaining sufficiently flexible to rapidly transition towards different cellular fates, are fascinating and provide powerful models to investigate the coordination of gene expression processes. We propose that the translational process is a major actor of gene expression regulation in stem cells, which not only precisely sustains a specific proteome required for maintaining undifferentiated cell identity and stem cell multipotent properties but also rapidly rewires gene expression in response to fate transition cues or environmental insults. Thus, most signaling pathways known to regulate stemness and differentiation also control global translational efficiency (Figure 1). Future progress in systems biology should allow a better prediction of translational responses in reaction to multiple signaling inputs.

In addition to global translational regulation, stem cell properties also rely on a complex network of genes undergoing specific translational regulation that imply specific mRNA cis elements, such as uORFs or IRESs. Although not discussed herein but documented in many other reviews, many trans factors, including RNA-binding proteins (e.g., Musashi) or non-coding RNA (i.e., microRNA and lncRNA), regulate gene-specific translation and occupy an important part in the global control of stem cell-specific proteomes. Thus, translational control of stem cell identity and changes in fate require the complex integration of comprehensive translational regulatory inputs as well as gene-specific translational regulation, which comprise an extremely complex network of cis- and trans-acting elements. In addition, ribosome biogenesis and homeostasis play a crucial role in the translational regulation network controlling stem cell identity and proved to be central regulators of cell fate transitions as well as central hubs coordinating multiple gene expression processes and cellular programs (Figure 2). 

Finally, we presented emerging new concepts suggesting that ribosomes may carry out regulatory functions unlike previously appreciated (Figure 3). Although both “ribosome concentration” and “specialized ribosome” models are supported by increasing evidence, future investigations should address whether the new ribosome functions are broadly implicated in gene expression regulation in physiological conditions and examine the global impact of such regulations on specific gene expression programs. The emerging plasticity of the translational machinery, both at the protein and RNA levels, provides a remarkable landscape of molecular variations possibly impacting ribosomal activity. Identifying the molecular machineries supporting these different levels of plasticity of ribosomes and establishing whether these levels are coordinated to generate different populations of functionally divergent ribosomes are two fascinating and challenging questions, which will likely animate the field of translation over this decade.

Finally, we propose that uncovering mechanisms governing translational control and ribosome biogenesis in stem cells represents a promising approach to investigate the contribution of these processes to pathological cancer stem cells (CSCs) and to design new therapeutic molecules targeting CSC translational regulatory pathways, CSC-specific ribosome biogenesis processes, or “specialized” CSC ribosomes.

## Figures and Tables

**Figure 1 cells-09-00497-f001:**
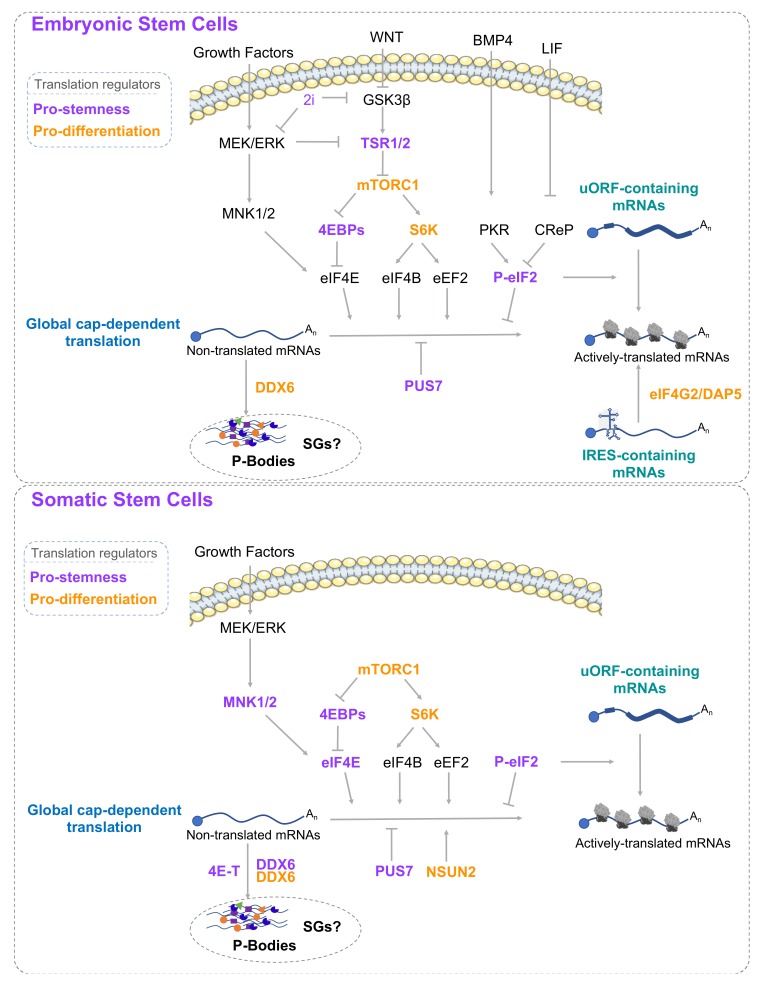
Translational control in stem cells. Upper panel: main signaling pathways that may regulate translational efficiency in embryonic stem cells. Translation regulators that promote stemness or in contrast stimulate differentiation are depicted in purple or orange, respectively. Lower panel: Similar to the upper panel in somatic stem cells. Translation regulators that promote stemness or induce differentiation are depicted in purple or orange, respectively. Images of cell membranes were kindly provided by the SMART Servier medical art database [9].

**Figure 2 cells-09-00497-f002:**
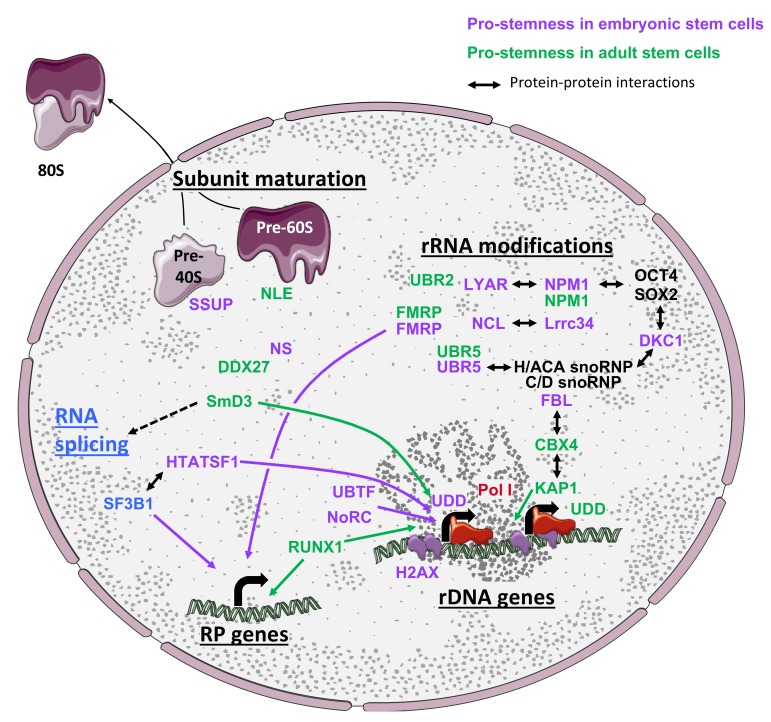
Ribosome biogenesis is highly regulated in embryonic and adult stem cells. Key RBFs described in this review and their implication in ribosome biogenesis in stem cells are presented. Factors stimulating ESC and ASC maintenance are depicted in purple and green, respectively. Direct protein/protein interactions are indicated by double-headed black arrows. Images of nucleic acids, the nucleus, and proteins were kindly provided by the SMART Servier medical art database [9].

**Figure 3 cells-09-00497-f003:**
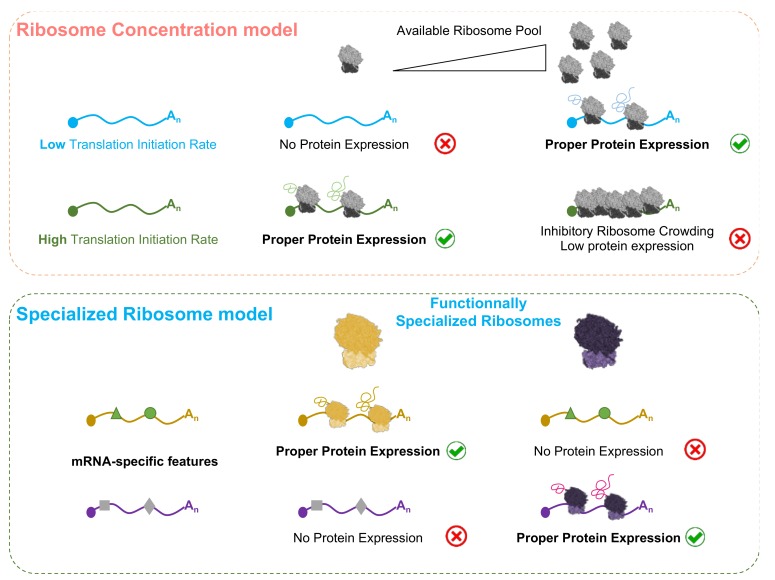
Ribosomes regulate gene-specific expression. Schematic diagram illustrating two models describing the impact of ribosome concentration on translation of specific mRNA subsets (upper panel) and the concept of functionally specialized ribosomes, which consists in heterogeneous ribosomes endowed with substrate selectivity (lower panel).

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
