# Peer review of "Ribosome and Translational Control in Stem Cells"

_cells, 2020, doi:10.3390/cells9020497_

Round 1
Reviewer 1 Report
The authors have provided a rather comprehensive summary on the stem cell regulation by ribosome biogenesis and translation, and the underlying mechanisms ranging from the regulations of signalling pathways (e.g. mTOR and MAPK) to the modifications of single factors (e.g. eIF4E and rRNA) in multiple stem cell systems. This review will attract a broad spectrum of readers from academia and industry who will largely benefit from it.
Ribosome biogenesis and translation are two highly coordinated/ coupled processes. The authors collect numerous references to demonstrate their roles in stem cell fate transition in part. 2 and part. 3, respectively. In the part. 3, many examples of how specialized ribosomes through either specific rRNA modification or ribosomal protein substitution may lead to qualitative control of gene expression, thus causing ribosomopathy when deregulated. Together with the part. 2, the mechanisms underlying global and selective translation outputs were elaborately discussed. In contrast, global ribosome biogenesis was only mentioned briefly in the model of ribosome crowding. It would be largely appreciated if there is some room for the up-to-date knowledge on how ribosome biogenesis is translationally mediated, e.g. how the translation of terminal oligopyrimidine tract mRNAs are mediated, and the relevance to stem cell fate decision making.
Minor points:
Line 459, …the control ‘of’ cell identity…; Line 820, …snoRNAs can ’be’ tissue specific…Author Response
We would like to thank the reviewer for the enthusiasm and interesting suggestions on our manuscript. We would like to mention that formatting errors of our manuscript removed sub-sections and may have created confusion on the general organization of the manuscript. We have corrected these mistakes and hope it will make it more comprehensible for the reader.
As suggested by the reviewer, we added a section discussing the regulation of RBF expression in stem cells. More precisely, we now discuss how mRNAs harboring 5’ Terminal OligoPyrimidine tract (TOP), which includes translation factors and RPs, are regulated in stem cells and during cell fate transition (part 3.5, 673-691). In addition, we modified part 4.1 to clarify observations that support the model of “ribosome concentration” and to discuss TOP mRNA expression regulation in stem cell in this context.
Minor point:
We corrected mistakes pointed out by the reviewer.
Reviewer 2 Report
The review manuscript by Gabut et al focuses on the posttranscriptional regulation of gene expression in stem cells. This is an interesting and timely topic. The authors do a good job providing a concise overview of the known signaling pathways converging on the regulation of translation efficiency. Figures in the manuscript are very nice an visually appealing. However, the review feels unbalanced in that some conceptually important areas in the existing literature are inexplicably ignored, while a whole lot of discussion is devoted to ribosome heterogeneity, which can be certainly a hook for those seeking a 'hot' topic, yet currently has few rigorously proven connections to stem cell biology. There are also some parts that feel like a collection of observations, often contradictory, with no satisfying discussion or hypotheses provided at the end. More work is needed before this manuscript is published in a form that would make it a useful resource for the readers.
Major issues.
1. The need to rapidly rewire gene expression is stated several times in the article. In this regard, altering ribosome concentration or the synthesis of specialized ribosomes seem like an incredibly slow way to regulate gene expression since it requires transcription of rRNAs and a multitude of ribosome components and assembly factors, snoRNAs, translation of mRNAs for ribosomal proteins, all in all a major shift in the cellular proteome on its own. This strikes me as inconsistent with the initial thesis. Can the authors provide some better rationalization for shifts in ribosome biogenesis as a regulatory mechanism that stem cells may require? I am also not convinced that there is no simpler explanation, namely, ribosome biogenesis genes simply become activated as part of increasing cellular biomass in a cell committed to division and downregulated when the stem cell is in a quiescent state.
2. A large body of literature exists on the linkage between defects in ribosome biogenesis and depletion of stem cell and progenitor populations through ribosomal/nucleolar stress and activated p53. Some key primary results, for example, can be found in (doi:10.1182/blood-2010-07-295238), (doi:10.4161/cc.11.3.19002), (doi:10.1111/j.1365-2141.2010.08396.x). I am also surprised that landmark findings such as from the Guy Sauvageau group (doi:10.1073/pnas.1418845112) with regard to differential effects of ribosomal protein deficiencies on ESC self-renewal vs differentiation are not even mentioned anywhere in the text. Omission of this information for a review of this kind is unacceptable. I understand that the authors want to emphasize new trends, but their case feels rather hollow without this foundation. Moreover, if the authors think that recently proposed models fit the observations better, please discuss the flaws in the old models and what concrete advantages are offered by the newer theories.
3. The authors need to clearly state the hypothetical conjectures as such in the text. For example, the Mills and Green model implies different ribosome concentrations affecting translation of specific transcripts (670-671). While reading this, one wonders if there any specific evidence linking ribosome abundance with differential gene expression or cell fate decision in stem cells. The lack thereof is only briefly mentioned at the end of the section (730). To avoid confusion, the authors need to clearly identify parts of their discussion based on speculative ideas throughout the manuscript. Regarding tissue specific effects in ribosomopathies, the authors should mention additional theories and cite relevant reviews on this subject, such as (doi:10.1242/dmm.020529) , (doi:10.1016/j.febslet.2014.03.024). See also Fig 3b in the excellent recent review by Nahum Sonenberg's group (doi:10.1038/s41580-018-0034-x). It would also be appropriate to cite the latter review somewhere in the text as an in-depth discussion of the translational deregulation.
4. Much of the discussion on ribosome heterogeneity does not seem to be relevant to stem cell biology. For example, in section 3.13, the first reference to a stem cell-related effect is in the second to last line (792). It is also unclear if the cited differences in ribosomal protein stoichiometry between monosomes and polysomes is a good argument for ribosome heterogeneity between different cell types: Wouldn't all cells display these differences (growth signaling-dependent, as the cited article seems to indicate), or do the authors propose that the monosome/polysome dichotomy with regard to ribosome composition is somehow different between stem and somatic cells? Is there any evidence in support of such a theory?
5. Line 580 – The cited study shows that Npm1 is linked to ribosomal RNA 2′-O-methylation, not rRNA maturation as a whole (that is, ribosomes can still be formed in cells with deficient Npm1 functions). Posttranscriptional methylation is just one aspect of rRNA maturation, but the two terms are not interchangeable. The authors do make this distinction in other places, but they should make sure they use various terms correctly throughout the text.
Minor issues.
123-137 PUS7 is first mentioned in the context of promoting differentiation, and in the next instance, as an inhibitor of differentiation. This is confusing; it would be better to combine this information in one place and provide a more coherent explanation of the PUS7 KO phenotype.
140 and 146 the idea of maintaining a proper translation balance is repeated twice in the same paragraph, one instance would be sufficient.
832 - "functionally specialized" is a conjecture at this point and should be removed from the section title. The authors themselves correctly note the speculative nature of the idea in the next paragraphs, for example lines 856-57, 872-873.
While the language of the review is clear, a few typos and grammatical inconsistencies are present, warranting another, and perhaps more thorough, spelling/grammar check. Below are a few typos, but this list is by no means is complete.
33 – missing noun after “have allowed” (researchers?)
180 participates 'in'
330 'disrupts'
459-464 formatting problems
597 'whose' loss is responsible
886 – the first sentence is too long; should be 'are' fascinating?
896 rely 'on'
Fig 1 – would the authors consider using the same color for “pro-stemness” in both panels? (the same goes for “pro-differentiation”). In my opinion, that would simplify comparing the two types of stem cells, I personally find a four-color scheme a little too difficult to process.
Fig 2 – Why do the authors call the proteins shown in this figure “main RBFs”? These are not implicated in major nuclear steps of ribosome biogenesis, contrary to what the figure legend says, but largely rRNA modifications.
Author Response
First of all, we would like to thank the reviewer for all comments and suggestions that will undoubtedly strengthen our manuscript. We performed changes as requested and detailed them below.
In addition, we would like to mention that formatting errors of our manuscript removed sub-sections and may have created confusion on the general organization of the manuscript. We have corrected these mistakes and hope it will make it more comprehensible for the reader.
General Comment:
To address the reviewer’s general comments, we added an entire section on the role of p53 pathway in stem cells (part 3.4.3, 628-656). Our goal was not to intentionally ignore major works, but we initially thought this was not covering the main focus of this review centered on the role of ribosome and translational control in stem cells. However, we fully understand the reviewer’s point and corrected it accordingly.
In addition, it was actually our choice to dedicate an entire section on ribosome heterogeneity, however absolutely not for the sake of “hooking those seeking a hot topic”, but because it is an interesting emerging concept that is likely to draw the attention of readers outside from the ribosome/translation fields, and if confirmed in the coming years, may impact the way the researcher community foresees gene expression regulation in a more broad spectrum. First, ribosome heterogeneity is not mentioned in the title for ‘hooking’ potential readers and only represents a section out of a 22-page review. Moreover, we believe that some recent breakthrough papers, such as those from Barna’s Lab in ESCs or the recent paper from the Nielsen and Cavaillé Labs, bring new sets of evidence that cannot be ignored and pave the way for a new biology of ribosome certainly important for the field of stem cell and developmental biology.
Last, we attempted to provide interesting discussions throughout our manuscript, and we however understand that some parts may seem too descriptive or confusing. This is in part the consequence of a sparse literature that still lacks some degrees of maturity on the subject. Yet, as requested by the reviewer, we made additional efforts to provide the readers with conclusions and constructive discussions as much as possible (493-500, 517-518, 527-529 etc…).
Here are our point-by-point answers to reviewer’s comments:
Major Comments
1) In our manuscript, “the need to rapidly rewire gene expression” is mentioned in the context of translational regulation that affect global and gene-specific translation (line 47, 59-60, 697-700, 354-359, 960-964 etc…), mostly via changes in signaling pathways as described in part 2. This hypothesis is supported by Munoz et al. Mol Syst Biol 2011, Kristensen et al. Mol Syst Biol 2013 and Lu et al. Nature 2019, which suggest that most of changes in ESC and iPSC proteomes are caused by translational regulation, as discussed in the manuscript. As these translational regulations can occur via signaling pathways that (i) are under the control of pro-stemness and pro-differentiation factors (Wnt, BMP, FGF etc…) and (ii) are by definition quick and do not require de novo gene synthesis, we do think it could participate in early events of cell identity changes.
We do not suggest that changes in ribosome biogenesis participate in early differentiation events as the reviewer postulated. However, we suggest that ribosome biogenesis is important for maintaining cell identity and homeostasis (395-399, 436-440) or is differentially regulated as cell undergo changes in identity (409-411, 473-476). As the reviewer mentioned, we agree that ribosome biogenesis changes could instead participate in intermediate/late events or reinforce the establishment of newly acquired cell identity. Of course, this theory will require further investigations at this point.
We found few sentences (387-389 and 954-956) that may be misleading and corrected them to avoid confusion as the reviewer suggested.
2) As the reviewer suggested, we added an entire section on p53-mediated nucleolar stress and stem cells (part 3.4.3, 628-656). We apologize for the unintentional omission of the work from Guy Sauvageau and colleagues and corrected this mistake. We added this reference to our manuscript and we thoroughly discuss it in different sections (parts 3.4.3 and 4.1). We hope the reviewer will now feel that our manuscript is more achieved.
3) As the reviewer requested, we made some changes to clarify some hypothetical conjecture we make in the manuscript. In addition, we intensively modified part 4.1 as requested to emphasize that the impact of “ribosome concentration” model on stem cell physiology is still speculative to date, while we provided indirect observations linking “ribosome concentration” regulation and stem cell physiology. As suggested by the reviewer, we modified the first paragraph of part 4.1 to acknowledge previous reviews on ribosomopathies and mention alternative hypotheses supporting tissue-specific defects of ribosomopathies. In addition, we justify our choice of focusing on hypotheses supporting regulatory functions of ribosomes to stay consistent with the topic of our manuscript.
4) The “specialized ribosomes” hypothesis, which relies on observations that ribosome are heterogeneous and that this heterogeneity provides specific functions, is intensively debated. Although “much of our discussion on ribosome heterogeneity does not seem to be relevant to stem cell biology” as the reviewer noticed, we think this is important to provide the readers with key and historical observations that built up this model in order to better appreciated convincing results, potential caveats and technical challenges to prove the existence of such ribosomes. To our opinion, this discussion is essential for providing the readers with a constructive review of the most recent discoveries that support ribosome heterogeneity and functionally specialized ribosomes in Embryonic Stem Cells. We want to emphasize here the point that these landmark discoveries were made in stem models therefore we think they are totally relevant for this review.
Nowhere in the paragraph pointed out by the reviewer we use the ribosome heterogeneity between monosome and polysome to argue that ribosome heterogeneity exists between different cell types. We simply make the point that ribosome composition is heterogeneous in ESC populations. This statement may seem evident for translation experts, but it may be striking for non-specialist readers as many textbooks still refer to the ribosome as a constitutive and uniform molecular complex with an invariable stoichiometry. In this way, convincingly demonstrating that ribosome composition varies in cells, here ESCs, is a major point toward proving the existence of ribosomes with different regulatory functions. As the reviewer asked, there is no published evidence for ribosome composition differences between stem and somatic cells to our knowledge, so it remains an exciting question that will undoubtedly be addressed in future works.
5) We thank the reviewer for pointed out this mistake. We have corrected the manuscript throughout.
Minor Comments
1) 124: We mention that PUS7 KO increases global translation and causes differentiation defects in ESCs. 137: we mention that, similarly to ESCs, PUS7 KO increases global translation and causes HSC differentiation defects. These similarities are also highlighted in Fig. 1. We are not sure what the reviewer 2 is confused about. However, we modified the text to make it as clear as possible and avoid confusion.
2) We corrected as the reviewer requested.
3) As suggested by the reviewer, we added a question mark to acknowledge the speculative aspect of the title at this point.
4) We performed a more thorough spelling/grammar inspection and had the manuscript be double checked by our English Editing Department. We thank the reviewer for pointing out these typos.
5) Fig1: We made the color changes requested by the reviewer.
6) Fig2: We modified the legend as requested by the reviewer.
Round 2
Reviewer 2 Report
The authors have done an excellent job with this revision and thoroughly addressed all the comments. The review is much more complete now and reads well. As far as this reviewer is concerned, the manuscript is ready for publication.